

# Deep-learning tool for early identification of non-traumatic intracranial hemorrhage etiology and application in clinical diagnostics based on computed tomography (CT) scans

Meng Zhao[1,2,*], Wenjie Li[2,3,*], Yifan Hu[4], Ruixuan Jiang[2], Yuanli Zhao[2,5], Dong Zhang[2,5], Yan Zhang[2,5], Rong Wang[2,3], Yong Cao[2,5], Qian Zhang[2,5], Yonggang Ma[2,3], Jiaxi Li[2,3], Shaochen Yu[2,3], Ran Zhang[6], Yefeng Zheng[4], Shuo Wang[2,3] and Jizong Zhao[2,3]

[1] Neurosurgery Department, Capital Medical University, Beijing, China
[2] China National Clinical Research Center for Neurological Diseases, Beijing, China
[3] Beijing Tiantan Hospital, Capital Medical University, Beijing, China
[4] Tencent You Tu Lab, Tencent, Shenzhen, China
[5] Department of Neurosurgery, Beijing Tiantan Hospital, Capital Medical University, Beijing, China
[6] Affiliated Hospital of Shandong Jining Medical College, Jining, Shandong, China
* These authors contributed equally to this work.

Corresponding author
Jizong Zhao, zhaojizong@bjtth.org

## ABSTRACT

**Background:** To develop an artificial intelligence system that can accurately identify acute non-traumatic intracranial hemorrhage (ICH) etiology (aneurysms, hypertensive hemorrhage, arteriovenous malformation (AVM), Moyamoya disease (MMD), cavernous malformation (CM), or other causes) based on non-contrast computed tomography (NCCT) scans and investigate whether clinicians can benefit from it in a diagnostic setting.

**Methods:** The deep learning model was developed with 1,868 eligible NCCT scans with non-traumatic ICH collected between January 2011 and April 2018. We tested the model on two independent datasets (TT200 and SD 98) collected after April 2018. The model's diagnostic performance was compared with clinicians' performance. We further designed a simulated study to compare the clinicians' performance with and without the deep learning system complements.

**Results:** The proposed deep learning system achieved area under the receiver operating curve of 0.986 (95% CI [0.967–1.000]) on aneurysms, 0.952 (0.917–0.987) on hypertensive hemorrhage, 0.950 (0.860–1.000) on arteriovenous malformation (AVM), 0.749 (0.586–0.912) on Moyamoya disease (MMD), 0.837 (0.704–0.969) on cavernous malformation (CM), and 0.839 (0.722–0.959) on other causes in TT200 dataset. Given a 90% specificity level, the sensitivities of our model were 97.1% and 90.9% for aneurysm and AVM diagnosis, respectively. On the test dataset SD98, the model achieved AUCs on aneurysms and hypertensive hemorrhage of 0.945 (95% CI [0.882–1.000]) and 0.883 (95% CI [0.818–0.948]), respectively. The clinicians achieve significant improvements in the sensitivity, specificity, and accuracy of diagnoses of certain hemorrhage etiologies with proposed system complements.

**Conclusions:** The proposed deep learning tool can be an accuracy tool for early identification of hemorrhage etiologies based on NCCT scans. It may also provide more information for clinicians for triage and further imaging examination selection.

## INTRODUCTION

Spontaneous and nontraumatic intracerebral hemorrhage (ICH) represents a significant global health concern, with an annual incidence of 10–30 per 100,000 population. As the most devastating stroke subtype, ICH accounts for 15–20% of all strokes and is associated with substantial morbidity and mortality worldwide (*Qureshi, Mendelow & Hanley, 2009*; *Labovitz et al., 2005*). While hypertension is the predominant cause of ICH, a considerable number of cases stem from underlying macrovascular abnormalities, including arteriovenous malformation (AVM), aneurysm, cavernous malformation (CM), and Moyamoya disease (MMD) (*Qureshi, Mendelow & Hanley, 2009*; *Meretoja et al., 2012*; *Zhao et al., 2023*). Timely and accurate diagnosis, coupled with appropriate surgical interventions, can significantly reduce mortality and improve functional outcomes in ICH patients (*van Donkelaar et al., 2020*). Thus, rapid and precise identification of ICH etiology is crucial, as treatment strategies vary considerably depending on the underlying cause, particularly for vascular abnormalities (*Hemphill et al., 2015*; *Derdeyn et al., 2017*).

The optimal approach for early identification of macrovascular causes in non-traumatic ICH patients remains a subject of debate (*Cordonnier et al., 2018*). Although digital subtraction angiography (DSA) was the gold standard for detecting macrovascular abnormalities, its invasive nature poses inherent risks. Non-contrast computed tomography (NCCT) has emerged as the primary diagnostic tool in emergency settings for patients presenting with symptoms suggestive of hemorrhage, owing to its widespread availability, cost-effectiveness, and rapid acquisition capabilities (*Hemphill et al., 2015*). Following ICH identification *via* NCCT, computed tomography angiography (CTA) and magnetic resonance angiography (MRA) are typically recommended for detecting underlying vascular lesions (*Hemphill et al., 2015*). Early diagnosis of these vascular abnormalities can significantly impact clinical management and prognostic predictions in ICH patients. However, the high costs and limited accessibility of time-intensive magnetic resonance imaging (MRI), MRA, and CTA often restrict their application in the acute phase (*Demchuk, Menon & Goyal, 2016*). In clinical practice, particularly in underserved healthcare environments, it's rarely feasible for every ICH case to undergo emergency CTA or MRA examination (*Bekelis et al., 2012*). Moreover, there's a lack of clear guidelines for selecting patients for further angiographic imaging to identify etiology (*Hilkens et al., 2018*; *Cordonnier et al., 2010*).

While NCCT can potentially screen for macrovascular causes, identifying key features remains challenging. Enhancing the sensitivity, specificity, and accuracy of NCCT screening, particularly in primary etiology identification of ICH, could prove invaluable in

assisting physicians to make rapid, well-informed decisions regarding further angiographic imaging or immediate intervention (*Singh & Bhatia, 2019*). Recent advancements in artificial intelligence, specifically convolutional neural networks (CNNs), have demonstrated excellent performance in various clinical image-based recognition tasks, showing promise as a diagnostic strategy (*Ye et al., 2019*). Several deep learning algorithms based on CNN models have received approval from the US Food and Drug Administration (FDA) for medical image interpretation, further underscoring their potential in this field (*Ye et al., 2019*). Given the potentially less favorable outcomes that from delayed clinical management due to ICH etiology identification process, such as vasospasm and delayed cerebral ischemia. An accurate and timely deep learning model that could help clinicians reliably identify ICH etiology from NCCT scans is especially valuable. However, to date, the integration of human and artificial intelligence (AI) for ICH etiology detection have barely started, and the ability of deep learning systems to augment clinician performance remains relatively unexplored. In the past, the subtype of ICH was predicted based on the 2D CNN model. *Ye et al. (2019)* tried to apply a 3D CNN-based approach to detect ICH, using a simple CNN network with five convolutional layers and two fully connected layers. The performance of this plain 3D CNN seems to improve. It is unclear whether this method can produce reliable predictions (*Ye et al., 2019*).

This study aimed to investigate the potential of a CNN system for diagnosis of ICH etiology from NCCT scans. The main hypothesis of our research is that a deep learning model can achieve high accuracy in classifying different ICH etiologies and provide significant diagnostic support to clinicians in a real-world setting. In addition, we also compared the performance of the proposed system with clinician diagnoses. We further conducted a simulated study to compare the clinicians' performance with and without the deep learning system complements.

## MATERIALS AND METHODS

### Datasets and clinical taxonomy

#### Data collection

For dataset development, we conducted a retrospective review of NCCT scans from 4,019 consecutive patients with non-traumatic acute ICH who presented at Beijing Tiantan Hospital between January 2011 and April 2018 (Fig. 1). Two radiologists independently confirmed the ICH diagnoses (*Meretoja et al., 2012*). We included only the initial NCCT scan following each ICH event in our analysis. Data were collected similarly as previously described by *Meretoja et al. (2012)* We also uploaded our detailed methods to Protocol.io (dx.doi.org/10.17504/protocols.io.yxmvmejoog3p/v1). Portions of this text were previously published as part of a preprint (*Zhao et al., 2023*).

We excluded examinations that were: (1) diagnosed as traumatic, (2) associated with a history of brain surgery or external ventricular drain, (3) conducted 24 h or more after the first ICH event, (4) duplicated scans with rescan due to compromised image quality, initial head motion and metal artifacts arisen from finger and ear rings, (5) lacked a complete image series, or (6) were from patients under 4 years of age. We reviewed medical records
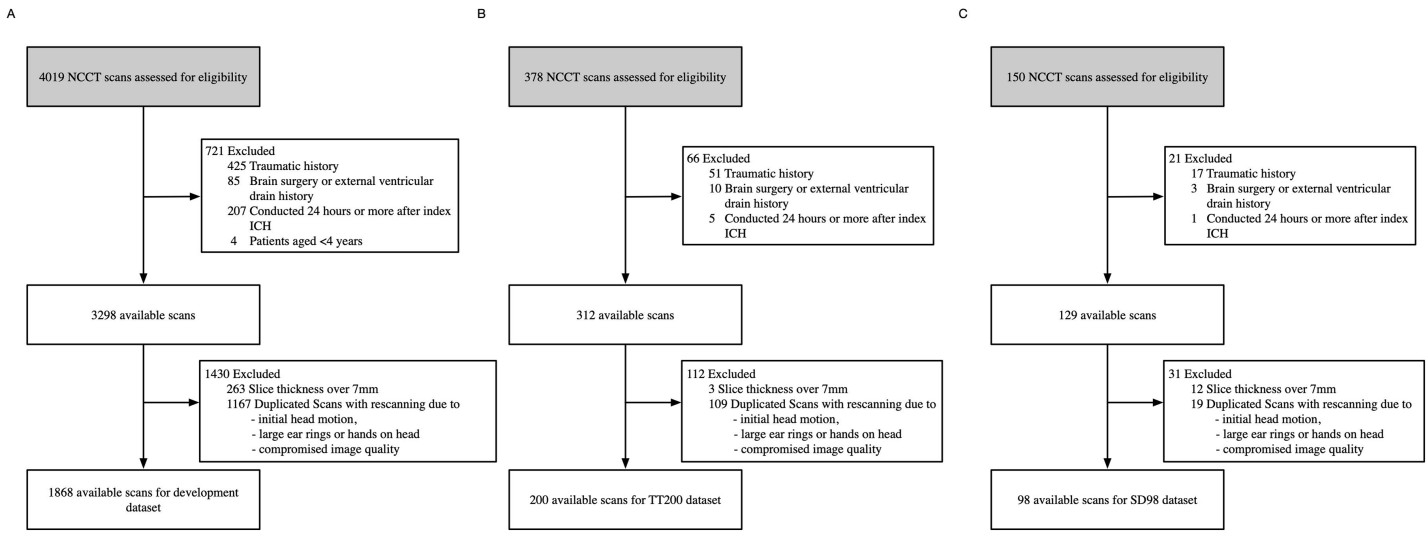

**Figure 1 Dataset selection flow chart.** (A) Development dataset. (B) TT200 dataset. (C) SD98 dataset.

for patient age, sex, complaint, medical history, known hypertension, and impaired coagulation (*Meretoja et al., 2012*). Of the original 4,019 records, 1,868 were eligible for the model development dataset.

To assess the accuracy and generalizability of our developed system, we collected two independent test datasets: NCCT scans of consecutive patients with non-traumatic ICH who presented at Beijing Tiantan Hospital from April 2018 to November 2018 (TT200), and NCCT scans of consecutive patients with non-traumatic ICH who presented at the Affiliated Hospital of Shandong Jining Medical College from April 2018 to January 2019 (SD98). The NCCT parameters were 120 kVp, 300 mA, 512 × 512 image matrix, 1-s rotation, 5-mm section thickness, 5-mm intersection space, and CTDIvol was approximately 40 mGy. We applied the same exclusion criteria, preprocessing, and labelling methods to both datasets as we did to the development dataset (*Meretoja et al., 2012*).

This study received approval from the Institutional Review Boards of the Beijing Tiantan Hospital and Affiliated Hospital of Jining Medical College (KYSQ 2019-163-01). All medical images and clinical data underwent complete anonymization (*Meretoja et al., 2012*). We retrieved Digital Imaging and Communications in Medicine (DICOM) images from picture archiving and communication system servers in compliance with the Health Insurance Portability and Accountability Act. To protect patient privacy, we removed all protected health information from the dataset, including names, dates of birth, medical record numbers, and other direct identifiers. We obtained written consent from all participants.

### Data labelling

Our etiologic classification and definitions were based on the method proposed by *Meretoja et al. (2012)*. We confirmed etiology diagnoses using NCCT scans in conjunction

with additional clinical evidence, including CTA, MRA, digital subtraction angiography, or surgical pathology notes. Two radiologists with over 10 years of experience labeled the ICH etiologies, with any disagreements resolved by a third investigator (*Meretoja et al., 2012*). We categorized each NCCT case as having one of the following six causes: hypertensive hemorrhage, aneurysm, AVM, MMD, CM, or other causes, which included cerebral amyloid angiopathy, systemic disease, undetermined causes, arteriovenous fistula.

## Data preprocessing and model development

### Data preprocessing

First, each volume was rotated every 20 degrees along the axial plane with a widely used bilinear algorithm (*Ballard & Brown, 1982*), resulting in 18 different image volumes. Second, each volume was resized into a new volume with a consistent physical voxel size of $0.6 \times 0.6 \times 4.2$ mm$^3$, which was almost the same as the mean voxel size of the training set (*Han, 2013*). Third, an unsupervised intensity-based skull-stripping algorithm was applied to the volume, and the stripped volume was cropped to $280 \times 280 \times 30$ voxels. We obtained a total of 33,624 file volumes for the training database.

### Model selection

The proposed ICHNet model demonstrated superior performance compared to other convolutional neural network architectures (AlexNet (*Krizhevsky, Sutskever & Hinton, 2012*), ResNet (*He et al., 2016*), and SENetNet (*Hu, Shen & Sun, 2018*)) across multiple classification tasks (See Table S1). ICHNet achieved the highest accuracy for hypertensive hemorrhage (0.9457), Moyamoya disease (0.8466), and cavernous malformation (0.8983). It ranked second in accuracy for aneurysm (0.9395) and arteriovenous malformation (0.8606) detection. Moreover, ICHNet tied with SENet for the highest pooled accuracy (0.7383) across all categories. While ICHNet did not achieve the highest accuracy for the 'Others' category, it still outperformed SENet in this metric (0.7108 *vs.* 0.6969). These results indicate that ICHNet exhibits robust capability in classifying various types of intracranial hemorrhages, often surpassing or matching the performance of established CNN architectures. AUCs and the overall accuracy served as the evaluation metrics (*Chilamkurthy et al., 2018*). The proposed model outperformed all of the previous models outperformed on most metrics (see Tables S1 and S3).

### Model development

We used ICHNet for classification of the causes of intracranial hemorrhages from NCCT scans which inspired by CNN architecture SlowFast Networks (*Feichtenhofer et al., 2019*) (see Fig. S1). The output of ICHNet was a vector for the probability of each cause of ICH. Using stratified random sampling, the initial dataset was split so that both the training and validation sets had similar proportions of the six class labels. Raw data/code is available DOI: 10.5281/zenodo.13337992.

## Training and test protocol

### Training procedure

We oversampled classes other than aneurysm and hypertension because of the imbalanced classes, with the training data for AVM, MMD, CM and others are repeated for six, 14, 17, and three times, respectively. The parameters of the model were initialized by the Kaiming method and optimized with an Adam optimizer (*He et al., 2016*). To find the optimal weights, the network was trained by the summation of two different losses, the weighted categorical cross entropy loss and the triplet loss (*Chechik et al., 2010*). The weights for the weighted cross entropy loss were computed based on the inverse frequency of each class before oversampling (Fig. S2). In the triplet loss function, the triplet margins of these four classes mentioned above were also half of the margins of the two main classes.

### Test procedure

An ensemble strategy was established to validate the proposed approach on the prospective test data by computing the average probability of five models from five-fold cross validation. The diagnosis could then be predicted from the average probability of all images of the same patient over AUCs. The training and testing procedures were implemented using the Pytorch-0.4.1 package with Python 3.6.

## Performance evaluation

Our model produced confidence scores (0–1) indicating the likelihood of each potential ICH etiology for given NCCT inputs. We applied these algorithms to each case in two independent test datasets. Diagnoses were assigned based on the highest predicted probability for each sample. To assess model robustness, we calculated overall accuracy and plotted receiver operating characteristic (ROC) curves for each diagnostic label. We evaluated the algorithms using accuracy, sensitivities, specificities, and area under the receiver operating curve (AUCs). Furthermore, we compared high-sensitivity and high-specificity points (both approximately 0.9) from the ROC curves for each label.

We applied the deep learning model to both test datasets and compared the results on the TT200 dataset with ICH etiology predictions from six expert raters: two neuroradiologists (with 7 and 21 years of experience) and four neurosurgeons (two with 6 years of attending experience, one with 1 year as attending, and one with 6 years as a resident). Importantly, none of these raters participated in data collection or model development.

The evaluation process comprised three distinct tasks for the raters:

Task One: Three neuroradiologists raters independently predicted ICH etiologies based solely on NCCT images.

Task Two: After a 14-day interval, all six raters made predictions using both NCCT scans and associated clinical data (age, sex, chief complaint, physical examination, and medical history).

Task Three: Following another 14-day period, raters made predictions with access to the model's probabilistic etiology predictions for each case. Raters could consider or disregard the model's confidence scores at their discretion.

Raters were informed about the model's performance metrics on the validation dataset but not on the test datasets. Each rater assigned one of six etiology prediction labels to each case, and all tasks were conducted without time constraints. This comprehensive evaluation process allowed us to assess both the model's performance and its potential impact on clinical decision-making when used as a supportive tool.

## Statistical analysis

For the model performance assessment, we generated confusion matrices for each of the six etiologies and plotted the true positive rate against the false positive rate for different possible thresholds in one-*vs.*-all diagnostic tests, and the AUC was calculated to evaluate the model. The AUC values for the model were calculated based on the model's prediction scores. The 95% confidence intervals (CIs) of sensitivity and specificity were then calculated using the exact Clopper-Pearson method based on the β distribution, *Clopper & Pearson (1934)* and 95% CIs of AUCs were calculated using the Hanley and McNeil method (*Hanley & McNeil, 1982*). The pooled accuracy was obtained from the diagnosis results of all raters. The concordance between paired raters was computed using Cohen κ coefficient (*Cohen, 1960*). The exact Fleiss κ was also obtained to measure the concordance of all raters (*Fleiss & Cohen, 1973*). To compare the performances of the algorithm and the raters, we applied the bootstrapping method to obtain samples of metrics for assessment. $p$-values of less than 0.05 were considered significant.

## RESULTS

For model development, we retrospectively reviewed the NCCT scans consecutive patients with non-traumatic acute ICH who presented at Beijing Tiantan Hospital between January 2011 and April 2018. Of 4,019 NCCT scans reviewed, 1,868 were eligible for inclusion, as shown in Fig. 1. For a qualified input NCCT DICOM volume of an ICH patient, the system was designed as an end-to-end approach, including skull-stripping, rotation and classification, that analyzed the entire NCCT volume automatically and produced a series of scores representing the probabilities of different etiologies. The TT1868 dataset was randomly divided into five folds with four folds in the training set and one-fold in the validation set; TT200 and SD98 were prepared as test sets (Fig. S2). The patient demographics and image characteristics from the 1,868 records used for model development are summarized in Table 1.

In our analysis of 1,868 ICH scans, we identified 628 (33.6%) as hypertension-related, 845 (45.2%) as aneurysm-related, 44 (2.4%) as MMD-related, 34 (1.8%) as CM-related, 104 (5.6%) as AVM-related, and 213 (11.4%) as related to other causes. Following pre-processing and data complementation, our dataset expanded to 33,624 NCCT volumes.

The TT200 dataset comprised 200 examinations: 75 hypertension-related, 70 aneurysm-related, 12 MMD-related, 11 AVM-related, 14 CM-related, and 18 related to other causes. The SD98 dataset included 61 hypertension-related, 25 aneurysm-related, five MMD-related, four AVM-related, one CM-related, and two other cause-related records (Table 1).

**Table 1 Characteristics for both training (TT1868) and two test datasets (TT200 and SD98).**

|  | Development dataset | TT200 | SD98 |
|---|---|---|---|
| Number of patients (with both scans and reports) | 1,868 | 200 | 98 |
| Mean age (SD; range) | 52.5 ± 14.9 (4–94) | 50.5 ± 15.9 (4–83) | 55.7 ± 13.0 (15–86) |
| Female patients | 805 (43.1%) | 86 (43.0%) | 37 (37.8%) |
| Etiologies |  |  |  |
| Aneurysm | 845 (45.2%) | 70 (30.0%) | 25 (25.5%) |
| Hypertensive hemorrhage | 628 (33.6%) | 75 (32.5%) | 61 (62.2%) |
| AVM | 104 (5.6%) | 11 (5.5%) | 4 (4.1%) |
| MMD | 44 (2.4%) | 12 (6.0%) | 5 (5.1%) |
| CM | 34 (1.8%) | 14 (7.0%) | 1 (1.0%) |
| Others | 213 (11.4%) | 18 (9.0%) | 2 (2.0%) |
| Undetermined | 116 (6.2%) | 5 (2.5%) | 0 |
| Cerebral venous thrombosis | 17 (0.9%) | 4 (6.2%) | 0 |
| CAA | 16 (0.9%) | 4 (6.2%) | 0 |
| AVF | 9 (0.5%) | 0 | 0 |
| Medication | 32 (1.7%) | 5 (6.2%) | 2 (2.0%) |
| Systemic disease/tumor | 23 (1.2%) | 0 | 0 |

**Note:**
AVM, Arteriovenous Malformation; MMD, Moyamoya Disease; CM, Cavernous Malformation; CAA, Cerebral Amyloid Angiopathy; AVF, Arteriovenous Fistula; Medication, Anticoagulation-related Medication.

Tables S1 and S2 present the comparative results of various CNN models for model selection. Table 2 and Fig. 2 summarize the performance of the ICHNet algorithm at selected operating points. On the TT200 dataset, ICHNet achieved AUCs of 0.986 (95% CI [0.967–1.000]) for aneurysms, 0.952 (95% CI [0.917–0.987]) for hypertensive hemorrhage, 0.950 (95% CI [0.860–1.000]) for AVM, 0.749 (95% CI [0.586–0.912]) for MMD, 0.837 (95% CI [0.704–0.969]) for CMs, and 0.839 (95% CI [0.722–0.959]) for other causes. On the SD98 dataset, ICHNet achieved AUCs of 0.945 (95% CI [0.882–1.000]) for aneurysms and 0.883 (95% CI [0.818–0.948]) for hypertensive hemorrhage.

Six raters evaluated the TT200 test dataset to assess the deep learning system's performance. In task one, based solely on image information, four clinicians achieved an average accuracy of 0.706, compared to the proposed approach's accuracy of 0.760. Bootstrapping tests indicated that our system significantly outperformed three of four clinicians ($p < 0.05$), as shown in Table S3 and Fig. S3.

In task two, with additional clinical information provided, raters achieved an average accuracy of 0.725. Despite this improvement, our approach still performed significantly better than three of six raters ($p < 0.05$) (Fig. S4 and Table S3).

In task three, with access to the algorithm's probability predictions, significant increases were observed in mean sensitivity to aneurysms (0.091, $p < 0.05$) and hypertensive hemorrhage (0.133, $p < 0.05$), and mean specificity to AVM (0.032, $p < 0.05$), MMD (0.044, $p < 0.05$), and CM (0.028, $p < 0.05$) with AI complements on NCCT images and clinical information. Table 3 and Fig. 3 present these results. Table S4 and Fig. S5 detail each rater's
**Table 2 Performance of algorithms on the TT200 dataset and SD98 dataset.**

| TT200 dataset | No. of positives | Number of negatives | AUC | High specificity point (=0.9): sensitivity (95% CI) | High sensitivity point (=0.9): specificity (95% CI) |
|---|---|---|---|---|---|
| Aneurysm | 70 | 130 | 0.986 (0.967, 1.000) | 0.971 [0.901–0.997] | 0.962 [0.913–0.987] |
| Hypertensive hemorrhage | 75 | 125 | 0.952 (0.917, 0.987) | 0.853 [0.753–0.924] | 0.840 [0.764–0.899] |
| AVM | 11 | 189 | 0.950 (0.860, 1.000) | 0.909 [0.587–0.998] | 0.910 [0.860–0.947] |
| MMD | 12 | 188 | 0.749 (0.586, 0.912) | 0.417 [0.152–0.723] | 0.468 [0.395–0.542] |
| CM | 14 | 186 | 0.837 (0.704, 0.969) | 0.571 [0.289–0.823] | 0.457 [0.384–0.532] |
| Others | 18 | 182 | 0.839 (0.722, 0.959) | 0.611 [0.358–0.827] | 0.528 [0.452–0.602] |
| SD98 dataset | | | | | |
| Aneurysm | 25 | 73 | 0.945 (0.882, 1.000) | 0.904 [0.795–0.952] | 0.920 [0.740–0.990] |
| Hypertensive hemorrhage | 61 | 37 | 0.883 (0.818, 0.948) | 0.649 [0.448–0.775] | 0.689 [0.540–0.787] |
| AVM | 4 | 94 | 0.872 | 0.553 | 0.750 |
| MMD | 5 | 93 | 0.796 | 0.172 | 0.800 |
| CM | 1 | 97 | 0.979 | 0.979 | 1.000 |
| Others | 2 | 96 | 0.781 | 0.740 | 0 |

**Note:**
AVM, Arteriovenous Malformation; MMD, Moyamoya Disease; CM, cavernous malformation; AUC, area under the receiver operating characteristic curve.

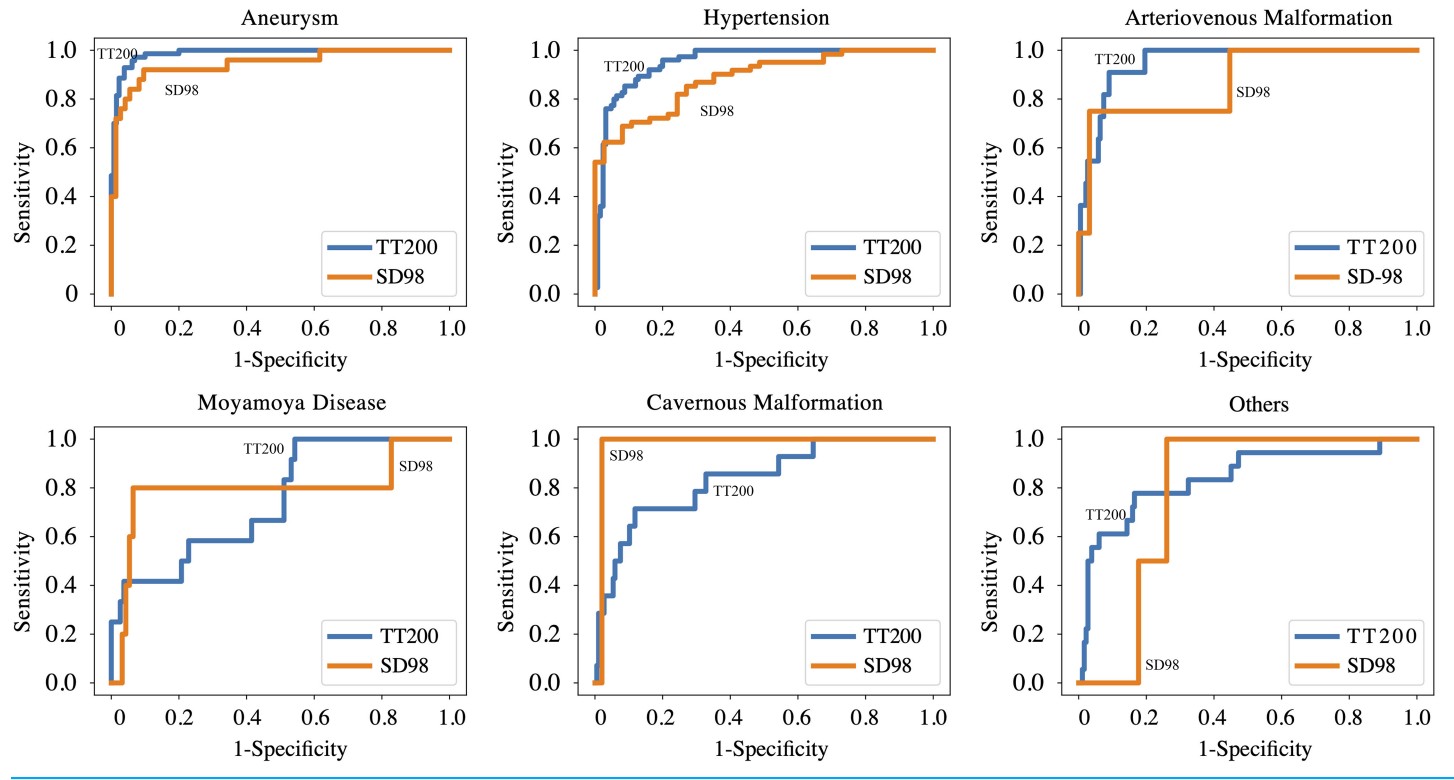

**Figure 2 Performance of proposed deep learning system on TT200 and SD98 test datasets.**

**Table 3 Clinician performance metrics with and without augmentation.**

| | Sensitivity | | | | Specificity | | | |
|---|---|---|---|---|---|---|---|---|
| | Without augmentation | With augmentation | Increment | *p*-value | Without augmentation | With augmentation | Increment | *p*-value |
| Aneurysm | 0.874 (0.791, 0.957) | 0.964 (0.944, 0.984) | 0.091 | 0.019 | 0.949 (0.933, 0.964) | 0.947 (0.938, 0.957) | −0.001 | 0.744 |
| Hypertensive hemorrhage | 0.798 (0.752, 0.843) | 0.931 (0.921, 0.942) | 0.133 | 0.002 | 0.917 (0.888, 0.946) | 0.896 (0.871, 0.921) | −0.021 | 0.926 |
| AVM | 0.697 (0.587, 0.807) | 0.712 (0.627, 0.797) | 0.015 | 0.500 | 0.920 (0.894, 0.946) | 0.952 (0.943, 0.960) | 0.032 | 0.018 |
| MMD | 0.264 (0.149, 0.379) | 0.236 (0.129, 0.343) | −0.028 | 0.690 | 0.943 (0.901, 0.986) | 0.988 (0.982, 0.993) | 0.044 | 0.008 |
| CM | 0.655 (0.588, 0.722) | 0.464 (0.300, 0.629) | −0.191 | 0.971 | 0.955 (0.943, 0.968) | 0.983 (0.972, 0.994) | 0.028 | 0.010 |
| Others | 0.222 (0.121, 0.324) | 0.343 (0.219, 0.466) | 0.120 | 0.144 | 0.979 (0.961, 0.997) | 0.973 (0.96, 0.987) | −0.006 | 0.836 |

Note:
AVM, Arteriovenous Malformation; MMD, Moyamoya Disease; CM, cavernous malformation; AUC, area under the receiver operating characteristic curve.

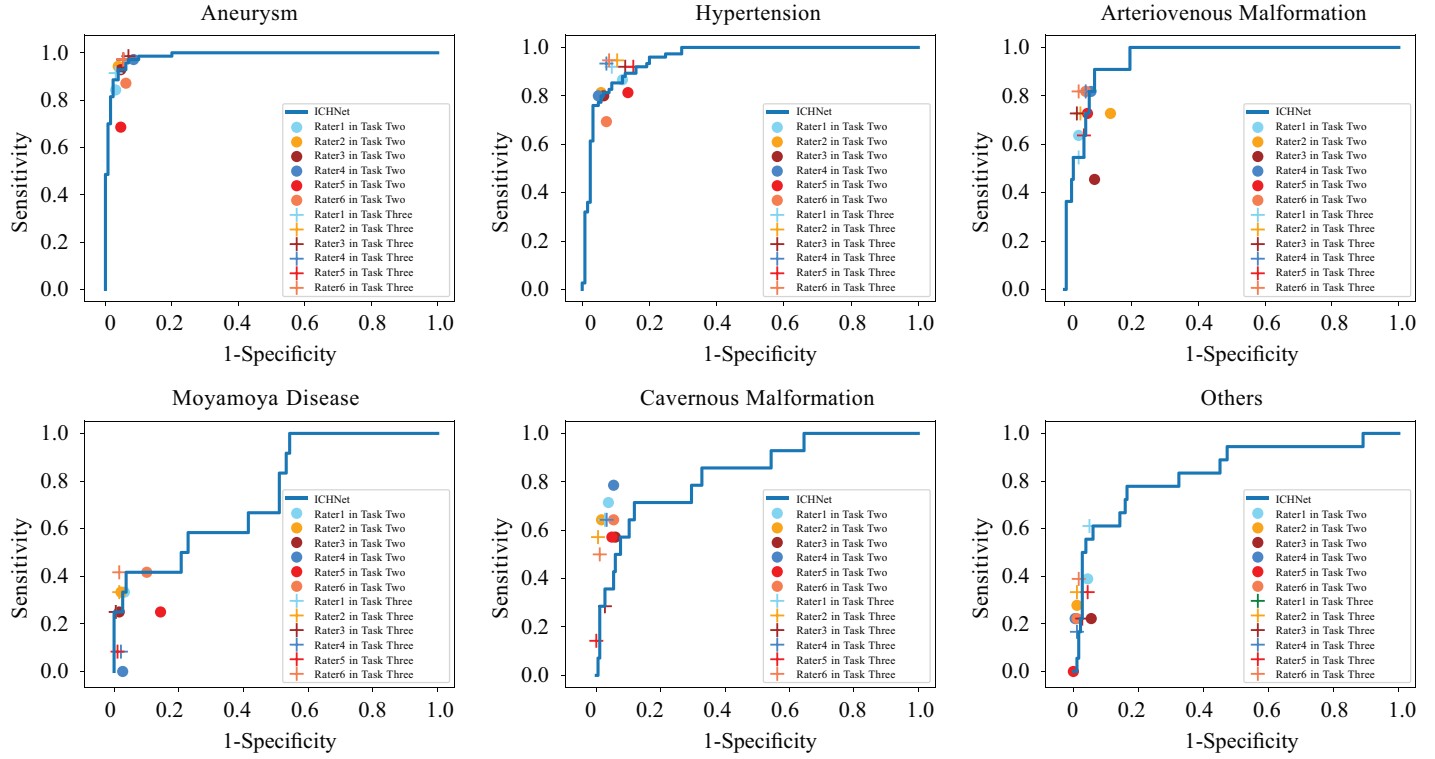

**Figure 3 Comparison between ROC curves of ICHNet's and clinicians' sensitivities and specificities before and after augmentation (Task Two and Task Three).** ROC, receiver operating characteristic.

performance. The mean accuracy of raters' diagnoses significantly improved from 0.725 to 0.803 (*p* < 0.01) with deep learning complements (Fig. S6).

Figure 4 presents the evaluated Cohen's kappa coefficients for pair-wise concordance. Pre-complement, these coefficients ranged from 0.51 to 0.69 across all rater pairs. Post-complement, concordance improved to 0.83 (*p* < 0.01), with nine of 15 pairs showing excellent agreement (>0.75) (*Fleiss, Levin & Paik, 2013*). The Fleiss' kappa for overall rater
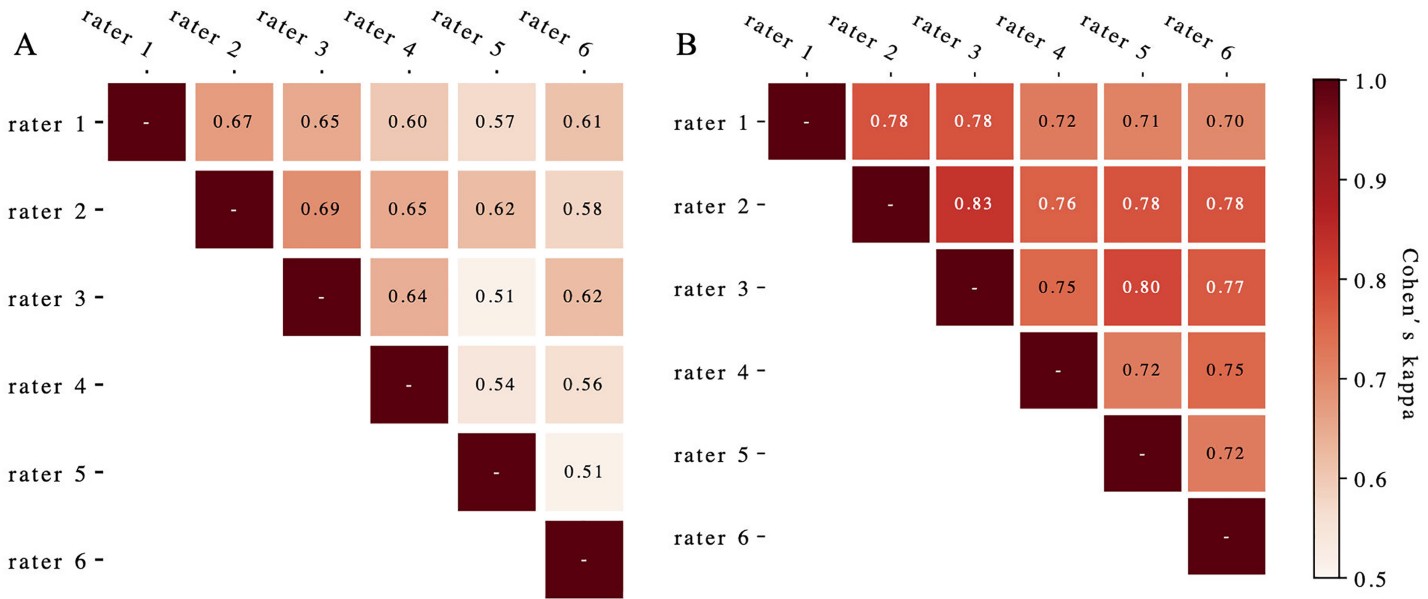

**Figure 4** The Cohen's kappa coefficients for pair-wise concordance before (A) and after (B) augmentation.

concordance increased from 0.61 to 0.75 ($p < 0.01$) after deep learning system complements.

## DISCUSSION

This study presents a pioneering deep learning system trained to diagnose ICH etiology from NCCT scans, achieving accuracy comparable to human experts. We validated the algorithm on two independent prospective datasets comprising diverse cases. Notably, when clinicians utilized the model's output, their diagnostic accuracy for ICH etiology significantly improved, surpassing the performance of the algorithm alone.

The field of AI in medical imaging has shown promising results, potentially enabling rapid, cost-effective, and accurate diagnostics with global accessibility. Deep learning algorithms have been successfully applied to various medical imaging modalities, including OCT scans (*De Fauw et al., 2018*), retinal fundus images (*Gulshan et al., 2016*), and digitized pathology slides (*Golden, 2017*). Previous studies have developed algorithms for identifying head computed tomography (CT) scan abnormalities requiring urgent attention (*Chilamkurthy et al., 2018*) and for detecting acute ICH and classifying its location from NCCT (*Lee et al., 2019*).

However, while progress has been made in applying deep learning to ICH CT imaging interpretation, identifying location alone is insufficient for clinical practice. Several prediction scores have been developed to assess the risk of underlying macro-vascular etiology in non-traumatic ICH patients, such as the simple ICH score, the secondary ICH score, and the DIAGRAM prediction score (*Hilkens et al., 2018*). Yet, these scores cannot predict or classify the specific ICH cause. Our system, capable of predicting various ICH
etiologies, could provide valuable information to clinicians for triage and imaging selection (*Singh & Bhatia, 2019*).

CT-angiography, although considered a standard diagnostic tool for ICH etiology, is not routinely performed in most centers (*Becker et al., 1999*). Studies on spontaneous ICH indicate that only 37.4–76.0% of ICH patients undergo CTA scans (*Bekelis et al., 2012*; *Oleinik et al., 2009*; *Sorimachi et al., 2020*). A large study involving 1,423 consecutive adult ICH patients reported CTA sensitivity of 95.7% for secondary ICH diagnosis (*Sorimachi et al., 2020*), with sensitivities of 99.1% and 90.4% for aneurysm and AVM diagnosis, respectively. Our algorithm demonstrated comparable performance on two independent datasets. At a 90% specificity threshold, our model's sensitivities in the TT200 dataset were 97.1% and 90.9% for aneurysm and AVM diagnosis, respectively, rivaling or potentially surpassing CTA performance.

To simulate a typical clinical setting, we provided raters with clinical information in the second task. Although this improved the sensitivity and specificity of clinicians' diagnoses, they generally remained less accurate than our proposed algorithm. Figure S8 illustrates cases where raters' diagnostic accuracy improved after considering the system's predicted probabilities. Figure S7 displays spatial resolution histograms for the training and test datasets, showing that the TT200 dataset's resolution distributions more closely resembled the training dataset (TT1868) than SD98.

When clinicians had access to the model's predicted probabilities, they achieved higher specificity in diagnosing AVM, MMD, and CM, while maintaining high sensitivity for AVM and MMD diagnoses. This is particularly crucial as false negatives in macrovascular etiology diagnoses should be minimized in clinical practice. Notably, the sensitivity of clinicians' diagnoses for hypertensive hemorrhage, the most common ICH etiology, increased significantly with AI assistance.

Our results demonstrate the model's high accuracy in classifying ICH etiologies, as evidenced by the impressive AUC values across various categories. The model's performance, particularly in challenging cases, underscores its potential to enhance clinical decision-making and optimize patient management strategies. It's crucial to emphasize that this deep learning system should be viewed as a supportive tool for clinicians rather than a standalone diagnostic device. Overreliance on AI systems can lead to issues such as automation bias or deskilling of clinicians. Therefore, we envision this tool as augmenting clinical decision-making, not replacing it.

This study had some limitations. Our study was hospital-based, and the two datasets were from two large tertiary referral hospitals. The distribution of the hemorrhage etiologies was unbalanced and may have been biased towards referrals. Despite the substantial number of scans with diversified etiologies, the number of cases in minority classes, like MMD and CM, were limited. While our results are promising, we acknowledge that clinical implementation of this system would require careful consideration and further validation. Future research should focus on developing and validating a protocol for integrating this tool into clinical workflows, with particular attention to patient safety and the potential for unintended consequences. Therefore, it is important to enrich the training database in the future work, especially the minority classes. Spectral CT provides enhanced

tissue characterization and improved contrast resolution, which could potentially augment the performance of our deep learning model (*Sedaghat et al., 2021*). Future investigations could explore how incorporating spectral CT data into our model might improve diagnostic accuracy and provide additional insights into various intracranial hemorrhage etiologies. Future work should also focus on developing explainable AI techniques for this system. Providing interpretable explanations for the model's predictions could enhance clinician trust, facilitate error detection, and potentially offer new insights into radiological features associated with different ICH etiologies.

## CONCLUSIONS

In conclusion, we presented a novel deep learning system that analyzes clinical NCCT scans and makes predictions of ICH etiology with sensitivities, specificities, and accuracies similar to those of clinical specialists. This system could potentially assist in triaging patients with ICH for further neurovascular evaluation, particularly in cases where the system indicates a high probability of etiologies such as aneurysm or AVM that would require immediate attention. We also demonstrated that integration of the deep learning model can augment clinician performance and could equip specialists with the ability to make better decisions. Further research is necessary to develop and validate a protocol for safely integrating this algorithm into clinical practice before assessing its potential impact on care outcomes.

### Funding

This work was supported by the National Natural Science Foundation of China (No. 81870904). The funders had no role in study design, data collection and analysis, decision to publish, or preparation of the manuscript.

### Grant Disclosures

The following grant information was disclosed by the authors:
National Natural Science Foundation of China: 81870904.

### Competing Interests

All authors declare that they have no competing interests. Yifan Hu and Yefeng Zheng are employed by Tencent You Tu Lab.

### Author Contributions

- Meng Zhao conceived and designed the experiments, performed the experiments, analyzed the data, prepared figures and/or tables, authored or reviewed drafts of the article, and approved the final draft.
- Wenjie Li conceived and designed the experiments, performed the experiments, analyzed the data, prepared figures and/or tables, authored or reviewed drafts of the article, and approved the final draft.

- Yifan Hu conceived and designed the experiments, performed the experiments, analyzed the data, prepared figures and/or tables, authored or reviewed drafts of the article, and approved the final draft.
- Ruixuan Jiang conceived and designed the experiments, performed the experiments, analyzed the data, prepared figures and/or tables, authored or reviewed drafts of the article, and approved the final draft.
- Yuanli Zhao conceived and designed the experiments, performed the experiments, analyzed the data, prepared figures and/or tables, authored or reviewed drafts of the article, and approved the final draft.
- Dong Zhang conceived and designed the experiments, performed the experiments, analyzed the data, prepared figures and/or tables, authored or reviewed drafts of the article, and approved the final draft.
- Yan Zhang conceived and designed the experiments, performed the experiments, analyzed the data, prepared figures and/or tables, authored or reviewed drafts of the article, and approved the final draft.
- Rong Wang analyzed the data, authored or reviewed drafts of the article, and approved the final draft.
- Yong Cao conceived and designed the experiments, performed the experiments, analyzed the data, prepared figures and/or tables, authored or reviewed drafts of the article, and approved the final draft.
- Qian Zhang conceived and designed the experiments, performed the experiments, analyzed the data, prepared figures and/or tables, authored or reviewed drafts of the article, and approved the final draft.
- Yonggang Ma conceived and designed the experiments, performed the experiments, analyzed the data, prepared figures and/or tables, authored or reviewed drafts of the article, and approved the final draft.
- Jiaxi Li conceived and designed the experiments, performed the experiments, analyzed the data, prepared figures and/or tables, authored or reviewed drafts of the article, and approved the final draft.
- Shaochen Yu conceived and designed the experiments, performed the experiments, analyzed the data, prepared figures and/or tables, authored or reviewed drafts of the article, and approved the final draft.
- Ran Zhang conceived and designed the experiments, performed the experiments, analyzed the data, prepared figures and/or tables, authored or reviewed drafts of the article, and approved the final draft.
- Yefeng Zheng conceived and designed the experiments, performed the experiments, analyzed the data, prepared figures and/or tables, authored or reviewed drafts of the article, and approved the final draft.
- Shuo Wang conceived and designed the experiments, performed the experiments, analyzed the data, prepared figures and/or tables, authored or reviewed drafts of the article, and approved the final draft.

- Jizong Zhao conceived and designed the experiments, performed the experiments, analyzed the data, prepared figures and/or tables, authored or reviewed drafts of the article, and approved the final draft.

## Human Ethics

The following information was supplied relating to ethical approvals (*i.e.*, approving body and any reference numbers):

The IRB of Beijing Tiantan Hospital, Capital Medical University granted Ethical approval to carry out the study within its facilities (KYSQ 2019-163-01).

## Data Availability

The data and code is available at GitHub: https://github.com/Tencent/MedicalNet.

The anonymised raw data measurements are available in the Supplemental File and Zenodo: Zhao. (2025). TTYY NCCT trainning and test dataset [Data set]. Zenodo. https://doi.org/10.5281/zenodo.14835185.

## Supplemental Information

Supplemental information for this article can be found online at http://dx.doi.org/10.7717/peerj.18850#supplemental-information.

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
