# Peer review of "Deep-learning tool for early identification of non-traumatic intracranial hemorrhage etiology and application in clinical diagnostics based on computed tomography (CT) scans"

_PeerJ, doi:10.7717/peerj.18850_

## Round 0.1 · original submission · Major Revisions

This article has now been reviewed by 3 experts, and they are recommending that major revisions are needed before it can be Accepted. Therefore, please address their comments in a revision and resubmit.

Reviewer 1 ·

Basic reporting

This study developed an AI system to accurately identify the causes of acute non-traumatic intracranial hemorrhage (ICH) using non-contrast CT scans. The model demonstrated high diagnostic performance and generalizability across independent datasets. Clinicians' diagnostic accuracy improved significantly with AI assistance, indicating the system's potential to enhance early identification of hemorrhage etiologies and support clinical decision-making. However, according to the guidelines for AI in medical imaging (Checklist for Artificial Intelligence in Medical Imaging (CLAIM): A Guide for Authors and Reviewers https://pubs.rsna.org/doi/10.1148/ryai.2020200029) I have a few concerns about this work. major revision is recommended before publication.

Experimental design

Abstract:
1. The author should indicate whether the data, and/or resulting model are available publicly.

Introduction
2. The author should provide a summary of the relevant literature of CNN and ICH and emphasize how this investigation expands upon and differentiates itself from previous studies.

Material and Methods:
3 Please include more information on CT scans: scan protocol(slice thickness, FOV, etc), manufacture(GE or Siemens or others)
4. The impact of scanner on the algorithms would be interesting. Please elaborate on the possible influence of machine vendor.
5.Three neurosurgeon should not rate independently. Either should they work with neuroradiologist to make consensus or all the rating should be done by neuroradiologist.
6. Detail the methods used to de-identify the data and remove protected health information in compliance with U.S. (HIPAA), European (GDPR), or other relevant regulations. Since facial profiles can enable identification, please specify the techniques employed to either remove or anonymize such information.
7. Describe the process by which the data were divided into training, validation, and testing sets. Specify the proportion of data allocated to each partition and provide justification for these choices. Additionally, indicate if there are any systematic differences between the data in each partition and explain the reasons for such differences, if applicable.
8.The demographic and clinical characteristics of cases in each partition should be clearly specified.

Validity of the findings

Result:
9. When ICH spills into the ventricular system, boundaries between ICH and intraventricular extension of intracerebral hemorrhage (IVH) are probably unclear, thus contribute to discrepancies in segmentation. How did the authors deal with this situation?
10. This is my primary concern regarding this study. The authors have only reported the AUC between the cohorts. For a comprehensive evaluation of AI performance, it is crucial to conduct a detailed quality analysis. Specifically, the authors should provide information on incorrect results and describe the nature of the network's errors. For example, an algorithm might achieve good segmentation of a hemorrhage but also incorrectly segment another object in the image while the overall similarity index might still be high.

·

Basic reporting

General comments:
1.1. The authors present an interesting deep-learning (DL) tool for early identification of non-traumatic intracranial hemorrhage (ICH) etiology based on non-contrast computer tomography (NCCT) scans and investigate whether clinicians can benefit from it in a diagnostic setting, however, the last part is not reflected in the title of the manuscript, while the tool application is considered widely and conclusions are focused on this.
1.2. The work is mostly well written, the English and structure are clear, the figures (except Figure 3) and tables, and the background/context with literature is sufficient and systematically introduced. However, there is a concern about the novelty of the methods. Many studies addressing similar methods/algorithms and modeling techniques can be found in the literature. As the focus is on methods applications, authors from the beginning should broaden the introduction about the approach of methods/tools‘ application and experiments‘ design which tends to reveal the improvement of diagnosis.
1.3. The manuscript is self-contained and with relevant results. How to estimate the performance of the deep-learning tool is quite clear. Still, the authors should explain why and on which theoretical basis (mentioning relevant references) they planned 6 raters and all three tasks, especially the last one, which is focused on the raters' predictions with the knowledge of the proposed tool.

Specific comments:
1.4. The results presented in the abstract are unclear. Please, firstly in the background part of the abstract clarify the classes of etiology and in the results part clarify the meaning of the statement “The model also shows an impressive generalizability in an independent dataset SD98”. The conclusions part just focuses on the application while the initial statement on accuracy (mentioned just in the background) could be also essential and related to the application.
1.5. In the abstract and conclusion, “triage and further imaging examination selection“ are not much considered in the manuscript, so at least the relevant discussion in the manuscript could be extended by providing possible procedures of the DL tool application.
1.6. In the introduction, the abbreviation DSA (and OCT in discussion) is not specified, and everywhere all references [x] are indicated not in the same sentence, but in the following sentence i.e. after the dot “.“ (that is unusual). Some parts of the discussion with a lot of references sound like the introduction, e.g. “Deep learning algorithms have been applied across a wide variety of medical scans ...“. It would be better to reorganize the discussion by moving some parts of a text to the background or introduction as some considerations are unclear or not related to the results, like the statement that “location identification alone is inadequate for clinical practice“.
1.7. Starting line 77, it would be good to extend the statement “ Given the potentially less favorable outcomes that from delayed clinical management due to ICH ...“ as this emphasizes the importance of manuscritp too. The vasospasm / delayed cerebral ischemia could be mentioned in this context.
1.8. In the result part, the wider and proper description of figures/tables could be useful (e.g. for Figure e-2, Table 1). Now there is an unprecise sentence “The Patient demographics and image characteristics from the 1868 records used for model development are summarized in Table 1“ or, in the discussion part, instead of Figure e-8 there is “ Figure e-7 showed illustrative cases that the raters’ diagnosis accuracy makes ...“.

Experimental design

General comments:
2.1. The research presented in the manuscript well reflects the aims and scope of the journal, but it would be good to know the main hypothesis and final research questions (considered in the discussion part and presented in the conclusions) at the beginning of the manuscript.
2.2. The investigation is rigorous and performed to a high technical & ethical standard, just would be good to present the theoretical basis of methodology that was followed for the considered case study (including a high variety of 6 raters and 3 tasks).
2.3. The methods described are clear and with sufficient detail & information to replicate, however, it is still unclear how the ground truth of ICH etiology is received. What is the gold standard and how it was considered? The dataset eligible for model development (together with the developed DL tool) still is not perfect like raters are not perfect, so for the generation of the confusion matrixes and accuracy estimation it is important to describe more widely the quality and labeling of data used for training of DL tool.

Specific comments:
2.4. In the data collection part, it is important to characterize the test dataset more clearly (including the initial and final sample size) and give references to relevant exclusion criteria.
2.5. The description of the data labeling (annotation) process should be extended and presented with additional tables or figures to present how many disagreements were considered and how statistically the labeling could be considered as true (all labels' correctness should be statistically significant) as this is directly important in DL model training and diagnosis accuracy estimation. As the basis, the given reference [3] indicates just that “etiology remained undetermined in 21%“.
2.6. In model selection Table e-1 indicates the same results for Pooled Accuracy (e.g. 0.7383 for SENet and ICHNet). Please check and explain how it could happen. The pooling could be based not only on the estimate of the mean but also on the median. Please also rename (e.g. by the number of parameters) or clarify the “Parameter Size“ in Table e-2.
2.7. The statement “The proposed model outperformed all of the previous models outperformed on most metrics (See Supplementary Table e-1 and e-3).“ is unclear while “previous models“ could be listed and “most metrics“ could be specified (defining the meaning of “overall accuracy“ and “Pooled Accuracy“).
2.8. More clarification in the results part could be useful, e. g. why “ only, four of the six clinicians participated in task one“. Also, „average accuracy“ could be provided in Supplementary material (Table e-3), Fleiss‘ kappa for the concordance among all raters could be presented like Cohen’s kappa coefficients, and Table e-5 somehow described too. Now there is an irrelevant indication of Table e-5 in the sentence “ Detailed performance of each rater was summarized in Supplementary Table e-4, Table e-5 and Figure e-5“, while no rater is considered there.

Validity of the findings

General comments:
3.1. The testing procedure of the DL tool is well described, and the validation process should be clarified at least at the same level. The annotation (of the six etiology class labels) procedure and validation dataset relation to clinical information as well as its preparation/augmentation and applications for the DL tool should be clearly described with relevant references to theoretical basis or a similar approach.
3.2. The data on which the results and conclusions are based are described quite well, but it would be good if they could be provided or made available in an acceptable discipline-specific repository. It is clear that medical images (DICOM) take a lot of memory but without them (just indicating the name) the dataset and labels of classes are useless. It would be nice if raw data could be shared as paper related database.
3.3. The conclusions could be elaborated more and related to the main results (quantitative estimates) concerning the 3 tasks and better decisions. The proposal for further research in the clinical setting could be quite risky as at first the new protocol should be proposed and validated. The statement “evaluation to exclude the presence of a vascular abnormality as the ICH etiology“ could be related just to binary classification and another type of labeling and DL tool. The single use of the DL system as a diagnostic tool could lead to various issues, which also could be mentioned in conclusions or at least discussions. The further application and importance of XAI (explainable artificial intelligence) could be also considered.

Specific comments:
3.4. Please clarify the meaning of the sentence „Using stratified random sampling, the initial dataset was split so that both the training and validation sets had similar proportions of the six class labels.“ Are you solving the issue of unbalanced classes or just adjusting validation sets? The training and test procedure could also be specified more clearly describing the mentioned “oversampling“ and “ensemble strategy“ concerning the validation or with a reference to Figure e-2.
3.5. Some terminology like “confidence scores“ (in performance evaluation) or „development dataset“ (in the manuscript or supplement material) is used as synonyms for other phrases, like “training dataset“. Please avoid possibly misleading synonyms and harmonize the whole manuscript.
3.6. In the performance evaluation part, please clarify or/and give references that support approaches related to the sentences “Each data sample was assigned diagnosis based on the largest predicted probability of the model“, “In addition, one high-sensitivity point and one high-specificity point from the ROC curves for each label were compared with one another“.
3.7. In line 179 of the manuscript please change “readers“ to “raters“. In Figure e-1 the box of label Conv2 could be blue instead of yellow and please check the titles of Figures e-7 (c) and (d) as they indicate “TT100“ instead of “TT200“.
3.8. The Figure e-7 is unclear (not described) and not considered in the manuscript. In line 280 there is the wrong reference to this figure instead of the right reference to Figure e-8.

Additional comments

4.1. The DL model/tool/system can “complement“ clinicians and similar statements in the whole manuscript (e.g. without/with the DL system) could be less misleading instead of the use of “augmentation“ terminology which in the DL field is more common for datasets.
4.2. In acknowledgment would be good to specify why thanks are expressed for the specific persons.

Reviewer 3 ·

Basic reporting

The authors developed an artificial intelligence system that can accurately identify
acute non-traumatic intracranial hemorrhage (ICH) etiology based on non-contrast CT and consecutively investigated whether clinicians can benefit from it in a diagnostic setting.
Interesting study, however, a minor revision should be considered.

Experimental design

Good.

Validity of the findings

Good.

Additional comments

Abstract: Please briefly describe the new AI algorithm you used, highlighting its significance in the field of medical imaging.
Introduction: Please clarify why your findings and the concept are novel compared to previously published work.
Methods: Please describe the novelty of the algorithm in more detail.
Discussion: The authors only used NCCT. However, CT nowadays often comes along with novel spectral CT technology. Please add a short paragraph or at least two sentences about future perspectives, emphasizing that your approach could be further investigated using spectral CT. Please add the following paper: DOI: 10.1055/a-1529-7010. Although it is not on ICH, the technique is the same, and it is one of the most recent works on that topic.

---

## Round 0.2 · Minor Revisions

As you can see, two of the reviewers have further comments which should be addressed in a revision.

Reviewer 1 ·

Basic reporting

Thank you for your response. While you’ve addressed the question to some extent, I still have some concerns.

Experimental design

Comment5:Neurosurgeons, while crucial in treatment, may lack the specialized expertise of neuroradiologists in interpreting NCCT images, potentially affecting diagnostic accuracy. To reduce bias and reflect real-world practice, I believe ratings should either involve neuroradiologists or be done in collaboration for more reliable results.

Validity of the findings

Comment 9: Sorry about the ambiguity of my previous comment. How do you ensure accuracy in distinguishing between intraventricular hemorrhage (IVH) and ICH with ventricular extension, given that these two conditions can easily be confused? What specific methods or criteria are used to differentiate their etiologies?

·

Basic reporting

Please consider/respond to the initial
General comments:
1.1. The authors present an interesting deep-learning (DL) tool for early identification of non-traumatic intracranial hemorrhage (ICH) etiology based on non-contrast computer tomography (NCCT) scans and investigate whether clinicians can benefit from it in a diagnostic setting, however, the last part is not reflected in the title of the manuscript, while the tool application is considered widely and conclusions are focused on this.
1.2. The work is mostly well written, the English and structure are clear, the figures (except Figure 3) and tables, and the background/context with literature is sufficient and systematically introduced. However, there is a concern about the novelty of the methods. Many studies addressing similar methods/algorithms and modeling techniques can be found in the literature. As the focus is on methods applications, authors from the beginning should broaden the introduction about the approach of methods/tools‘ application and experiments‘ design which tends to reveal the improvement of diagnosis.
1.3. The manuscript is self-contained and with relevant results. How to estimate the performance of the deep-learning tool is quite clear. Still, the authors should explain why and on which theoretical basis (mentioning relevant references) they planned 6 raters and all three tasks, especially the last one, which is focused on the raters' predictions with the knowledge of the proposed tool.

Experimental design

Please consider/respond to the the parts of the initial questions or
General comments:
...
2.2. ... just would be good to present the theoretical basis of methodology that was followed for the considered case study (including a high variety of 6 raters and 3 tasks).
2.3. ... What is the gold standard and how it was considered?

Specific comments:
2.4. In the data collection part, it is important to characterize the test dataset more clearly (including the initial and final sample size) and give references to relevant exclusion criteria.
...
2.8. More clarification in the results part could be useful, e. g. why “ only, four of the six clinicians participated in task one“. Also, „average accuracy“ could be provided in Supplementary material (Table e-3), Fleiss‘ kappa for the concordance among all raters could be presented like Cohen’s kappa coefficients, and Table e-5 somehow described too. Now there is an irrelevant indication of Table e-5 in the sentence “ Detailed performance of each rater was summarized in Supplementary Table e-4, Table e-5 and Figure e-5“, while no rater is considered there.

Validity of the findings

-

Additional comments

-

Reviewer 3 ·

Basic reporting

Clearly improved manuscript.

Experimental design

Good.

Validity of the findings

Good.

Additional comments

-

---

## Round 0.3 · accepted · Accept

The authors did a great job in responding to all the reviewers and the manuscript can be accepted for publication.

Reviewer 1 ·

Basic reporting

The authors fully addressed my concerns. I recommend acceptance.

Experimental design

no comment

Validity of the findings

The authors fully addressed my previous concerns.

·

Basic reporting

Thank you for your replies and updates on your manuscript. The manuscript now conform to professional standards of courtesy and expression. It include sufficient introduction and background to demonstrate how the work fits into the broader field of knowledge. The structure of the article conform to an acceptable format.

Experimental design

Research question well defined, relevant & meaningful. It is stated how research fills an identified knowledge gap.

Validity of the findings

The data on which the conclusions are based is provided and made available. The conclusions are appropriately stated and are connected to the original question investigated.

Additional comments

-